# Structure of catalase determined by MicroED

**Brent L Nannenga, Dan Shi, Johan Hattne, Francis E Reyes, Tamir Gonen***

Janelia Research Campus, Howard Hughes Medical Institute, Ashburn, United States

**Abstract** MicroED is a recently developed method that uses electron diffraction for structure determination from very small three-dimensional crystals of biological material. Previously we used a series of still diffraction patterns to determine the structure of lysozyme at 2.9 Å resolution with MicroED (*Shi et al., 2013*). Here we present the structure of bovine liver catalase determined from a single crystal at 3.2 Å resolution by MicroED. The data were collected by continuous rotation of the sample under constant exposure and were processed and refined using standard programs for X-ray crystallography. The ability of MicroED to determine the structure of bovine liver catalase, a protein that has long resisted atomic analysis by traditional electron crystallography, demonstrates the potential of this method for structure determination.

## Introduction

MicroED is an emerging method, which uses electron diffraction to obtain structural information from extremely small three-dimensional (3D) crystals of biological material. In the original MicroED proof of concept paper (*Shi et al., 2013*), electron diffraction data were collected from stationary lysozyme microcrystals and the structure was determined to 2.9 Å resolution from a series of still diffraction patterns. The method was substantially improved by employing 'continuous rotation' where data were recorded as the crystals were continuously rotated, resulting in higher quality data and allowing simple integration with existing processing programs used for X-ray crystallography (*Nannenga et al., 2014*). This led to the structure of lysozyme being determined to 2.5 Å resolution with improved statistics and data quality relative to the original lysozyme MicroED study.

In this work, we used thin bovine liver catalase 3D microcrystals for structure determination by MicroED. Catalase is a more difficult target than lysozyme, because it has a much larger unit cell, lower symmetry, and four molecules in the asymmetric unit. Moreover, each catalase monomer contains one heme group as well as a bound NADP molecule (*Fita and Rossmann, 1985*). Catalase is one of the earliest samples studied by EM, but despite extensive efforts spanning decades, the 3D structure of catalase has not been solved using electron diffraction. This is because the crystals have variable thicknesses (6–10 protein layers have been reported [*Dorset and Parsons, 1975b*]), and therefore these crystals were not suitable for 3D structure determination by traditional electron crystallography procedures (*Longley, 1967*; *Matricardi et al., 1972*; *Unwin, 1975*; *Unwin and Henderson, 1975*; *Dorset and Parsons, 1975a*). Here we report the 3.2 Å structure of catalase determined by MicroED. This is an important next step for the MicroED method as the analysis was rapid, taking a total of 3 weeks from crystal growth to final structural refinement, and used data from only a single catalase microcrystal.

## Results and discussion

### Sample preparation and data collection

Catalase was chosen for this study as it readily forms thin 3D microcrystals, which can be analyzed by transmission electron microscopy (TEM) (*Sumner and Dounce, 1937*; *Dorset and Parsons, 1975a*;

*For correspondence: gonent@
janelia.hhmi.org

**Competing interests:** The authors declare that no competing interests exist.

**Reviewing editor**: Stephen C Harrison, Harvard Medical School, United States

*Baker et al., 2010*). Well-ordered catalase microcrystals were grown by solubilizing crystalline catalase from an aqueous suspension followed by overnight dialysis in 0.05 M Sodium phosphate, pH 6.3. Following crystal formation, crystals were deposited on holey carbon grids and the sample was blotted and vitrified in liquid ethane prior to TEM analysis. The grids were screened in over-focused diffraction mode for the presence of thin 3D crystals (*Figure 1*, inset). The average crystal dimensions were found to be on the order of 8 µm by 4 µm in length and width and approximately 150 nm thick, in good agreement with previous crystallization results (*Dorset and Parsons, 1975b*). Suitable crystals were assessed by collecting a single still diffraction pattern at a total dose of 0.05 e⁻/Å², where well-ordered crystals showed sharp reflections extending to approximately 3.0 Å for untilted crystals (*Figure 1*). When high-quality diffraction was observed for an untilted crystal, the crystal was then tilted to 60° to check the diffraction quality at higher tilt angles as crystal flatness and embedding could affect the diffraction quality at higher tilt (*Gonen, 2013*). Typically, well-embedded and relatively flat crystals yielded data to ~2.8 Å resolution untilted but only ~3.2 Å at high tilt. Data sets were then collected as a sequence of 6 s exposures per frame. An example data set is shown in *Video 1* and was recorded as the stage was continuously rotated as described previously (*Nannenga et al., 2014*).

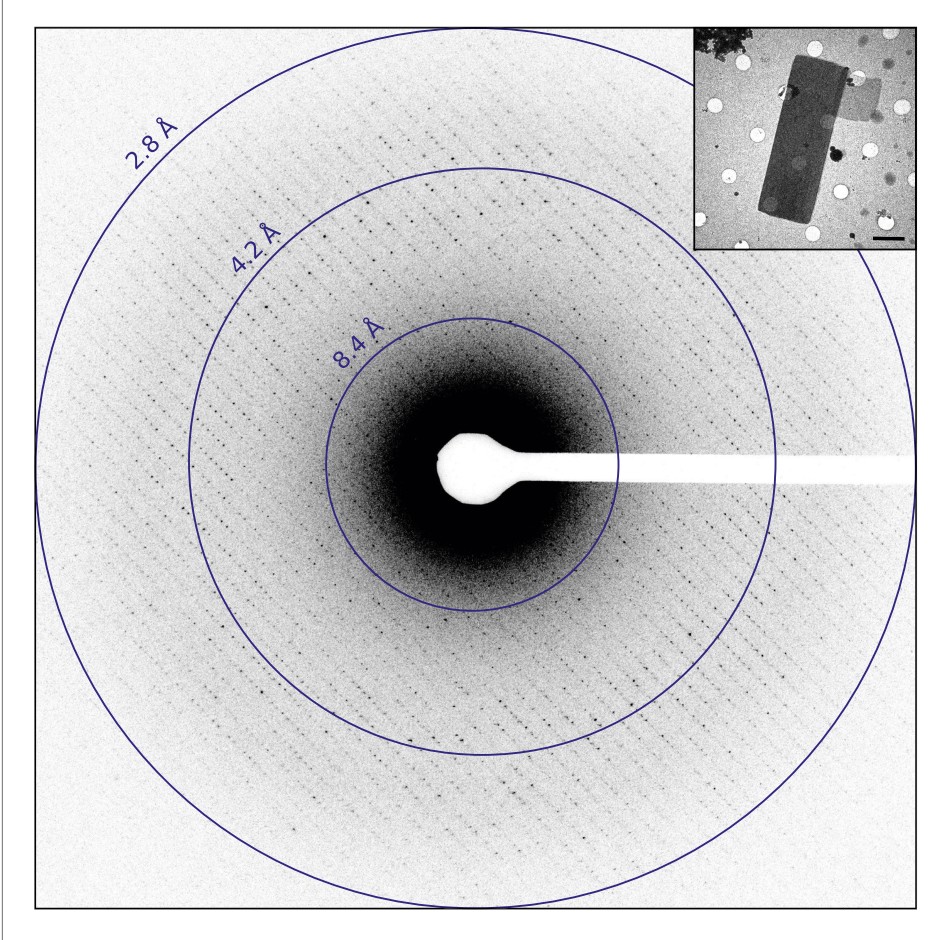

**Figure 1**. Diffraction from catalase microcrystals. Representative untilted still diffraction pattern of a catalase microcrystal that shows sharp reflections extending to approximately 3.0 Å. Crystals of this quality were used to collect a data set by continuous rotation. Inset shows an example catalase microcrystal as seen in over-focused diffraction mode. Scale bar is 2 µm. The dimensions of most microcrystals varied between 6 and 20 µm in length, 2 and 8 µm in width, and the thickness was approximately 100–200 nm, in agreement with previous catalase crystal sizes (*Dorset and Parsons, 1975b*).

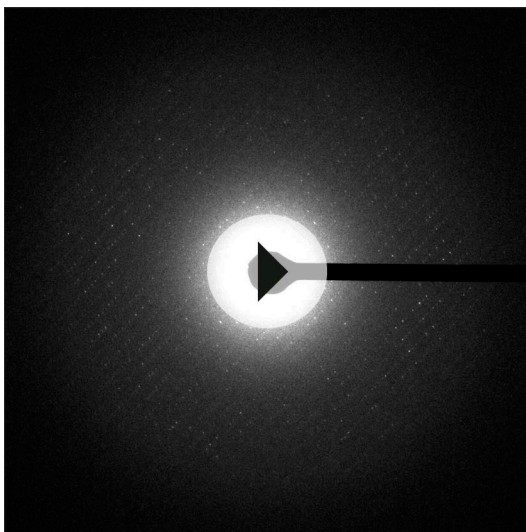

**Video 1**. Catalase diffraction data set collected by continuous rotation. Diffraction data was recorded at an exposure time of 6 s per frame from a single crystal as the stage was continuously rotating at ~0.09° s⁻¹.

## Data analysis, processing, and structure refinement

We sought to analyze the levels of dynamic scattering in our catalase data in order to validate whether kinematical scattering could be assumed. We quantified the dynamic scattering of the catalase crystals using the ratio of the strongest diffracted intensity to the unscattered incident beam (*Unwin and Henderson, 1975*), as well as the ratio of the sum of all diffracted intensities on an image to the incident beam intensity as described by *Dorset and Parsons (1975a)*. For the kinematical theory to apply, these ratios must be low. A representative crystal, which was approximately 200 nm thick, showed a ratio of $8.4 \times 10^{-3}$ for the maximum intensity and 0.18 for the sum of all intensities, which are close to the previously reported values (*Unwin and Henderson, 1975*; *Dorset and Parsons, 1975a*), indicating that kinematical assumption is valid for these diffraction data.

Data sets from five microcrystals were each integrated with MOSFLM (*Leslie and Powell, 2007*) followed by merging and scaling with POINTLESS (*Evans, 2011*) and AIMLESS (*Evans and Murshudov, 2013*). All crystals yielded comparable resolution but varied in data completeness (*Table 1*). We combined all five data sets in an effort to increase completeness, multiplicity and to improve the quality of the data and in parallel we processed the data from crystal 4 separately for comparison. Crystal 4 was chosen for processing separately because it had a good compromise between resolution and completeness. The data sets were processed to a range of resolutions (*Table 2*) and phases were determined by molecular replacement (MR) as implemented in MOLREP (*Vagin and Teplyakov, 1997*) with PDB ID: 3NWL (*Foroughi et al., 2011*) as a search model. Following MR, refinement using PHENIX and REFMAC (*Murshudov et al., 1997*; *Adams et al., 2010*) with electron scattering factors was performed. Merging data from multiple crystals did not have a significant effect on data completeness, because the catalase crystals all orient on the grid with the c-axis parallel to the electron beam (*Table 2*). When comparing the statistics presented in *Table 2*, it was clear that merging multiple crystals had a negative impact on the final refinement statistics, most likely due to non-isomorphism between crystals, and therefore the multiple crystal data sets were disregarded. Crystals 1, 2, 3, and 5 were relatively isomorphous and their

**Table 1.** Resolution and completeness for catalase data sets

|  | Crystal 1 | Crystal 2 | Crystal 3 | Crystal 4* | Crystal 5 |
|---|---|---|---|---|---|
| Resolution (Å) | 27.9–2.8 | 21.1–2.9 | 16.4–2.9 | 21.2–3.0 | 20.8–3.5 |
|  | (3.0–2.8) | (3.0–2.9) | (3.0–2.9) | (3.1–3.0) | (3.9–3.5) |
| Unit cell dimensions |  |  |  |  |  |
| a (Å) | 68.7 | 68.5 | 67.8 | 67.8 | 67.9 |
| b (Å) | 170.4 | 170.1 | 171.1 | 172.1 | 171.3 |
| c (Å) | 205.0 | 203.6 | 204.5 | 182.1 | 202.3 |
| α = β = γ (°) | 90 | 90 | 90 | 90 | 90 |
| Completeness (%) | 30.2 | 28.1 | 51.7 | 76.5 | 61.6 |
|  | (37.1) | (24.0) | (42.2) | (62.4) | (56.9) |

Values in parentheses reflect the highest resolution shell.
*Data set used for final structure.

**Table 2.** Comparison of merging and refinement statistics for catalase data sets

| | Multi-2.8 Å | Multi-3.0 Å | Multi-3.2 Å | Single-3.0 Å | Single-3.2 Å* |
|---|---|---|---|---|---|
| Total crystals | 5 | 5 | 5 | 1 | 1 |
| Resolution (Å) | 27.8–2.8 | 27.8–3.0 | 27.8–3.2 | 21.2–3.0 | 21.2–3.2 |
| | (2.9–2.8) | (3.1–3.0) | (3.4–3.2) | (3.1–3.0) | (3.4–3.2) |
| $R_{merge}$ (%) | 25.8 | 25.5 | 24.8 | 18.7 | 17.5 |
| | (42.0) | (41.2) | (38.2) | (38.8) | (32.7) |
| $CC_{1/2}$ | 0.906 | 0.920 | 0.933 | 0.886 | 0.891 |
| | (0.231) | (0.348) | (0.574) | (0.426) | (0.555) |
| Multiplicity | 4.8 | 5.2 | 5.6 | 2.4 | 2.4 |
| | (2.0) | (3.1) | (3.6) | (2.2) | (2.3) |
| Completeness (%) | 71.8 | 78.1 | 80.1 | 76.5 | 79.4 |
| | (35.2) | (61.8) | (72.2) | (62.4) | (75.5) |
| Mean ($I/\sigma(I)$) | 3.2 | 3.4 | 3.5 | 3.2 | 3.4 |
| | (2.5) | (2.2) | (2.1) | (1.7) | (2.0) |
| $R_{work}/R_{free}$ | 36.1/39.2 | 37.1/38.4 | 34.6/37.3 | 27.3/31.9 | 26.2/30.8 |

Values in parentheses reflect the highest resolution shell.
*Data set used for final structure.

diffraction data merged well, but the completeness of this data set was low (~62%), and we chose not to use it.

The single crystal 4 dataset was used for the remainder of the study. Close analysis indicated that the information content in the 3–3.2 Å resolution shell was too low for inclusion, which was not surprising as the crystals did not diffract well beyond 3.2 Å at high tilt angles. We therefore truncated the resolution to 3.2 Å yielding a final model with acceptable refinement statistics ($R_{work}/R_{free}$ = 26.2%/30.8%) and geometry (*Table 3*, *Figure 2A*, *Video 2*). The final $2mF_{obs}$-$DF_{calc}$ density map shows well-defined density surrounding the final refined model (overall map CC = 92.3%), both around the backbone and the side-chains (*Figure 2B*, *Video 2*), without significant peaks in the $mF_{obs}$-$DF_{calc}$ difference density map (*Figure 2C*). Additionally, the solvent channels between the tetramers in the crystal lattice show very little density (*Figure 2D*), further evidence of the quality of the model and data. The final structure of catalase at 3.2 Å resolution determined by MicroED agrees well with previously solved X-ray structures, with an RMSD of 0.358 Å and 0.440 Å between the MicroED structure and PDB ID: 3NWL (*Foroughi et al., 2011*) and PDB ID: 4BLC (*Ko et al., 1999*), respectively.

## Model validation

The orientation of our crystals prevented the full sampling of reciprocal space leading to systematic incompleteness (missing wedge or missing cone). While the incompleteness of data is significant, the resulting maps are still expected to be good enough for proper interpretation (*Glaeser et al., 1989*). To test the quality of the data and the resulting maps, and to identify any significant model bias or negative effects of the data incompleteness, the data were put through several validation tests. First, the robustness of the MR solution was tested by repeating the MR with a single monomeric chain of PDB ID: 3NWL (*Foroughi et al., 2011*) instead of the complete tetramer that was used originally. Even with a single chain, a strong solution was found with all four molecules successfully placed to recreate the complete tetramer (*Figure 2—figure supplement 1*; top MOLREP contrast score = 35.8, where a score of >3.0 is considered a strong solution. An identical test with synchrotron X-ray data yielded a top score of 23.7). We also phased the data with a poly-alanine model derived from PDB ID: 3NWL, and the resulting maps show clear density beyond the model where the correct side-chains could be rebuilt (*Figure 2—figure supplement 1B,C*).

In order to test the quality of the resulting maps following MR, autobuilding with Buccaneer (*Cowtan, 2006*) was performed and yielded 2120 residues in 160 fragments (*Figure 2—figure supplement 1D*).

**Table 3.** Data collection and refinement statistics

| Data Collection | |
|---|---|
| Excitation voltage | 200 kV |
| Electron source | Field emission gun |
| Wavelength (Å) | 0.025 |
| Total dose per crystal (e−/Å$^2$) | ~6.8 |
| Frame rate (frame/s) | 1/6 |
| Rotation rate (°/s) | 0.09 |
| Angular range per frame (°/s) | 0.54 |
| No. crystals used | 1 |
| Total angular range collected (°) | ~61 |
| **Merging Statistics*** | |
| Space group | $P2_12_12_1$ |
| Unit cell dimensions | |
| a, b, c (Å) | 67.8, 172.1, 182.1 |
| α = β = γ (°) | 90 |
| Resolution (Å) | 21.2–3.2 (3.4–3.2) |
| Total reflections | 67,064 (9873) |
| $R_{merge}$ (%) | 17.5 (32.7) |
| Total unique reflections | 28,143 (4278) |
| Multiplicity | 2.4 (2.3) |
| Completeness (%) | 79.4 (75.5) |
| Mean (I/σ(I)) | 3.4 (2.0) |
| $CC_{1/2}$ | 0.891 (0.555) |
| **Data Refinement** | |
| Reflections in working set | 26,732 |
| Reflections in test set | 1369 |
| $R_{work}$/$R_{free}$ (%) | 26.2/30.8 |
| RMSD bonds (Å) | 0.006 |
| RMSD angles (°) | 1.05 |
| Ramachandran (%)† | |
| (Favored, allowed, outlier) | 96.6; 3.3; 0.1 |

*Values in parentheses reflect the highest resolution shell.
†Statistics given by MolProbity (**Chen et al., 2010**).

Out of the built residues, 1280 traced the correct backbone and 467 side-chains were correctly assigned. Manual curation in Coot (**Emsley et al., 2010**), exploiting the fourfold non-crystallographic symmetry, resulted in a nearly complete model indicating the maps initially produced from our data were of good quality.

The next validation test performed involved removing sections of the model and analyzing the effect this had on the resulting refined density maps. This was done in order to examine the strength of the data and to find any potential model bias introduced by the MR search model. For this test, two validation models were used, in which the same sections of all four monomers of the final tetrameric structure were removed. The first model had residues 181 to 185 removed (Δ181–185) and the second model lacked the four heme groups (Δheme) that are normally found in catalase. Following refinement and simulated annealing, the resulting difference maps from the Δ181–185 model (**Figure 3A**) and Δheme model (**Figure 3C**) both showed significant positive difference density corresponding to the deleted regions of the model. Additionally, automated ligand identification was performed on the Δheme maps using phenix.ligand_identification (**Terwilliger et al., 2006**, **2007**), and the program was able to correctly place two of the four heme groups present in the structure. The results of these tests indicate that the data do not suffer from bias introduced by the MR search model.

Next, we sought to determine whether data from MicroED was of sufficient accuracy to locate small molecule ligands and corresponding protein conformational changes in the ligand-binding pocket. Bovine catalase binds four NADP cofactors, one per monomer through several side-chain interactions including F197 (**Kirkman and Gaetani, 1984**; **Fita and Rossmann, 1985**). Recently, the structure of bovine catalase was solved lacking the NADP cofactor (PDB ID: 3RGP) (**Purwar et al., 2011**), and in the NADP-free structure F197 underwent a conformational change as it no longer interacted with NADP. The crystals used for our MicroED analysis do contain NADP. Therefore, we used PDB ID: 3RGP (NADP-free structure) as a molecular replacement model against the MicroED data to determine whether NADP could be visualized in our catalase crystals using difference maps. When analyzing the difference maps, positive density was observed in the location expected for NADP although it appeared fragmented even at lower contour levels (**Figure 3E,G**). For visual comparison, structure factors from PDB ID: 3NWL, which was solved by X-ray crystallography, were truncated to 3.2 Å, and difference maps were calculated for Δ181–185, Δheme and NADP-free model (**Figure 3B,D,F,H**). Maps from both MicroED and synchrotron X-ray diffraction appear fragmented around the NADP even at lower contour levels (**Figure 3G,H**). The difference maps for both the MicroED and X-ray synchrotron data suggest that F197 should change its orientation to assume its correct position for NADP binding.

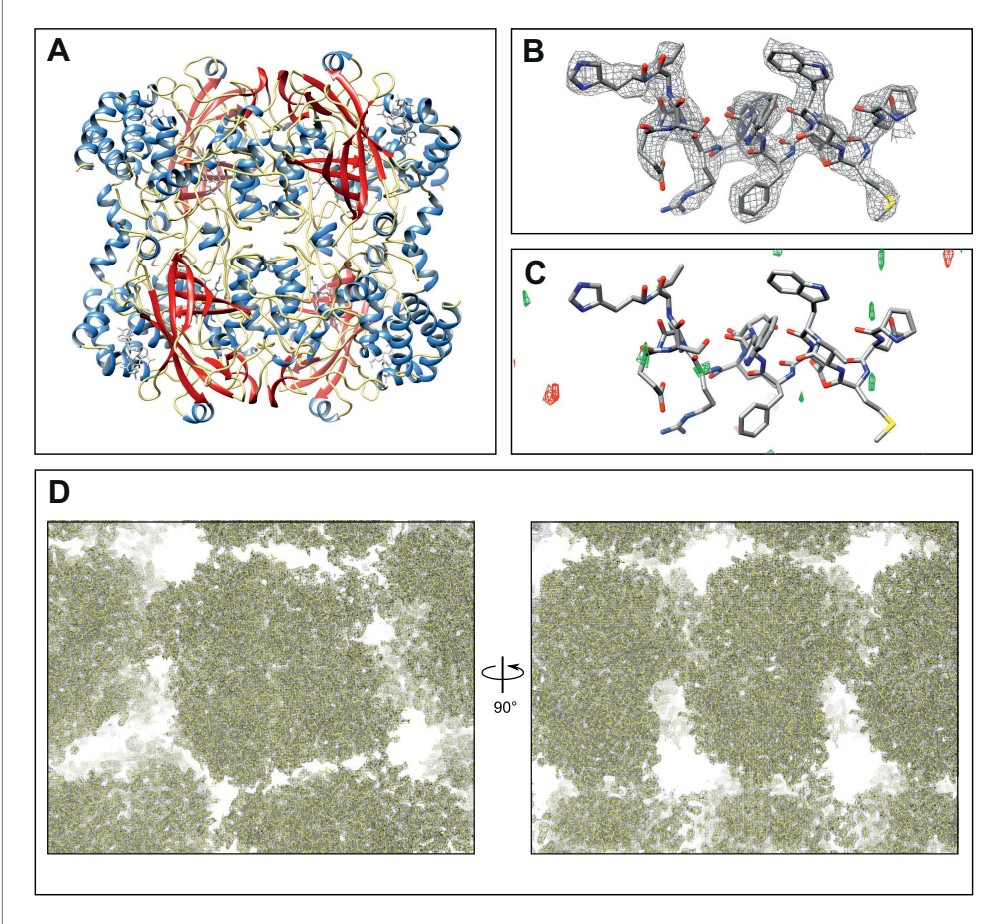

**Figure 2**. Final structure of catalase at 3.2 Å resolution determined by MircoED. (**A**) The complete refined catalase structure shown as ribbons. The corresponding $2mF_{obs}$-$DF_{calc}$ density map around the complete structure can be seen in **Video 2**. (**B**) The density map, contoured at 1.5 $\sigma$, around a representative region of the structure (residues 178–193 of chain **A**) shows well-defined density around the model. (**C**) The $mF_{obs}$-$DF_{calc}$ difference map, contoured at +2.5 $\sigma$ (green) and −2.5 $\sigma$ (red), around the same region shows no interpretable densities near the model indicating no obvious differences between the observed data and the calculated model. (**D**) Views of the density map (contoured at 1.0 $\sigma$) around a large region of the crystal lattice shows there is no significant density in the in the solvent channels of the catalase crystal.

The following figure supplement is available for figure 2:

**Figure supplement 1**. Molecular replacement validation tests.

These results demonstrate the MicroED data is of sufficient quality to detect subtle differences among structures at atomic resolution. At the current level of methodology with samples that suffer from missing data like catalase, MicroED produces lower quality maps than synchrotron X-ray diffraction. However, the catalase crystals used for MicroED were approximately 1000 times smaller in volume than those used at the synchrotron (**Foroughi et al., 2011**), and the resulting maps are still of high enough quality to determine the structure.

## Concluding Remarks

We present here the second protein structure determined by the emerging MicroED method. Bovine liver catalase resisted structural determination by traditional electron crystallography for decades, but the structure was readily determined by MicroED in 3 weeks from crystal formation to final structure determination using a single crystal. This is the second example where a single crystal was sufficient for structure determination by MicroED (**Nannenga et al., 2014**). Moreover, the continuous

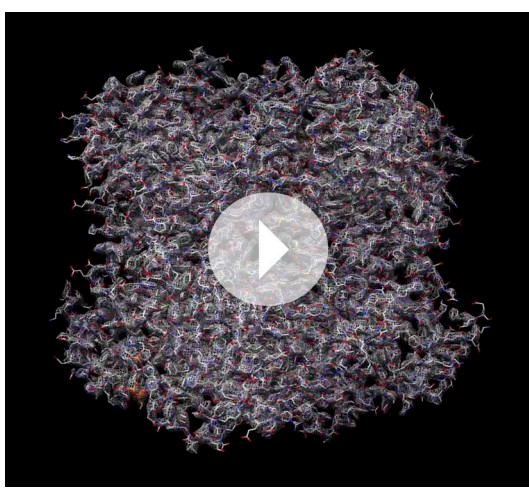

**Video 2**. Final density map and model of catalase at 3.2 Å. The $2mF_{obs}-DF_{calc}$ density map contoured at 1.5 σ shows good agreement with the final refined model.

rotation method yields data similar in quality to X-ray diffraction allowing simple processing with existing X-ray data reduction software and further accelerating structure analysis by MicroED (*Nannenga et al., 2014*). The resulting maps allow us to distinguish between subtly different protein conformations and to identify of small-molecule ligands such as NADP. This study shows that MicroED can be used as an alternative to X-ray crystallography using extremely small crystals for both mechanistic studies as well as structure-based drug design studies where small ligands are assayed.

# Materials and methods

## Catalase crystallization and sample preparation

Catalase was recrystallized from a commercial aqueous suspension of catalase (C100; Sigma–Aldrich, St. Louis, MO) by first centrifuging the crystalline suspension and dissolving the pellet in 1.7 M NaCl. The solubilized catalase was then centrifuged and the supernatant was dialyzed against 50 mM sodium phosphate pH 6.3 overnight at 4°C. Crystals were removed from dialysis, stored in an Eppendorf tube, and incubated an additional 24 hr at 4°C. Catalase crystals were stored at 4°C and were washed with water prior to sample preparation. To prepare samples for the TEM, crystals were resuspended and the undiluted catalase crystal suspension was applied, blotted and vitrified in liquid ethane as described previously (*Shi et al., 2013*).

## Collection of electron diffraction data

All electron diffraction was performed on a FEI Tecnai F20 TEM operated at 200 kV with a selected area aperture (6 µm in diameter at the specimen) and data were collected with 4k × 4k TVIPS F416 CMOS cameras (15.6 µm pixel size). Diffraction data were collected with a frame rate of 1 frame per 6 s as the sample was continuously rotated from high to low tilt angle at ~0.09° s$^{-1}$ (0.54°/frame) as described previously (*Nannenga et al., 2014*). A data set of approximately 61° was collected from a single crystal. Crystal thickness was estimated by measuring the intensity of the crystal (I) relative to the intensity of a hole in the carbon film ($I_0$) from an image and using Beer's law:

$$ln\frac{I}{I_0} = -\varepsilon ct$$

where $\varepsilon$ is the molar absorptivity, $c$ is the molar concentration, and $t$ is the crystal thickness. As an approximation, the value of $\varepsilon c$ for catalase was assumed to be the same as those for calculated for lysozyme. The lysozyme coefficients were determined using images of lysozyme microcrystals with a known thickness as described previously (*Nannenga et al., 2014*).

## Data processing and structure refinement

Raw TEM diffraction data were converted and processed using MOSFLM v7.1.0 (*Leslie and Powell, 2007*) and it's graphical interface iMOSFLM v1.0.7 (*Battye et al., 2011*), POINTLESS (*Evans, 2006*), and AIMLESS (*Evans and Murshudov, 2013*) as described in previous work (*Nannenga et al., 2014*). MOLREP (*Vagin and Teplyakov, 1997*) was used to perform molecular replacement using catalase PDB ID: 3NWL (*Foroughi et al., 2011*) as a search model (MOLREP contrast score = 40.8), and the molecular replacement solution was refined in PHENIX (*Adams et al., 2010*) and REFMAC (*Murshudov et al., 1997*) using a 5% free data set. Maps in *Figure 3E,F,G and H* were calculated using BUSTER-TNT (*Blanc et al., 2004*). Maps and models were displayed using the UCSF Chimera package (*Pettersen et al., 2004*).

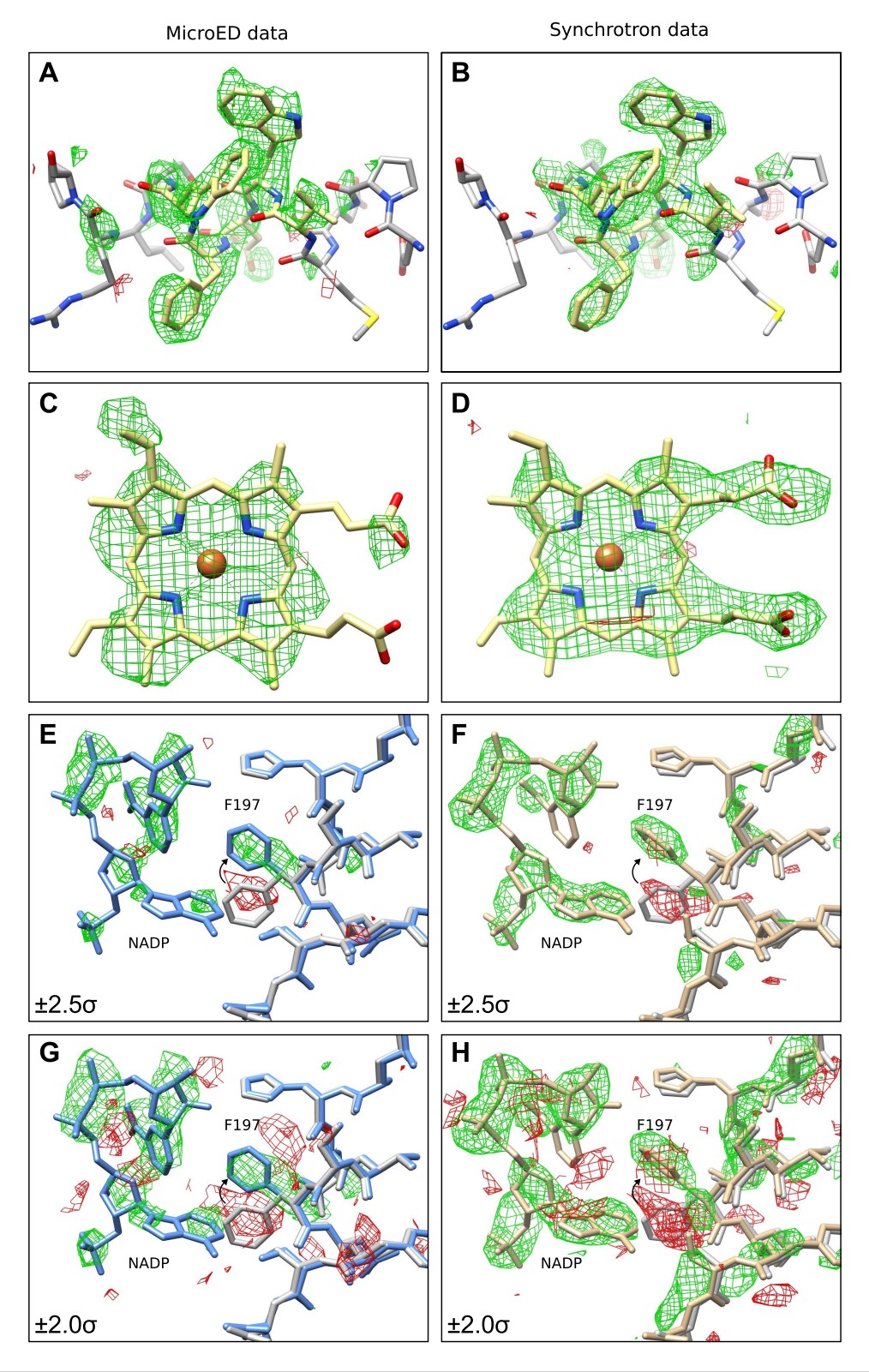

**Figure 3**. Final Model validation tests. (**A–D**) To check for model bias in the final MicroED data (**A** and **C**), and to compare with data obtained from synchrotron X-ray diffraction (**B** and **D**), residues 181–185 (**A–B**) or the heme groups (**C–D**) from all four chains were removed prior to simulated annealing and re-refinement. In all cases, significant positive $mF_{obs}$-$DF_{calc}$ difference density (contoured at ±2.0 $\sigma$) can be seen for the missing regions, which

*Figure 3. Continued on next page*

*Figure 3. Continued*

are displayed in yellow for clarity, indicating no significant model bias in the final structure. (**E**–**H**) When a model (PDB ID: 3RGP; gray) lacking NADP was used to phase the MicroED data or the synchrotron x-ray data, both of which contained NADP, significant density in the $mF_{obs}$-$DF_{calc}$ difference map was observed although it appeared fragmented even at a lower contour level. This density could be fit with the NADP and the conformational change of residue F197 present in our final refined model is apparent. Maps presented in panels (**E**–**H**) are all unrefined, following MR.

## Acknowledgements

The authors wish to thank Garib Murshudov (MRC LMB) for providing a version of REFMAC with support for electron scattering factors and Andrew Leslie (MRC LMB) for data processing support and advice. We also would like to thank Steven Sawtelle (HHMI Janelia Research Campus) for technical support and Joanita Jakana (Baylor) for the protocol for catalase crystallization. Work in the Gonen lab is supported by the Howard Hughes Medical Institute.

## Additional information

### Funding

| Funder | Author |
|---|---|
| Howard Hughes Medical Institute | Brent L Nannenga, Dan Shi, Johan Hattne, Francis E Reyes, Tamir Gonen |

The funder had no role in study design, data collection and interpretation, or the decision to submit the work for publication.

### Author contributions

BLN, DS, TG, Conception and design, Acquisition of data, Analysis and interpretation of data, Drafting or revising the article; JH, FER, Analysis and interpretation of data, Drafting or revising the article

### Author ORCIDs

Johan Hattne, http://orcid.org/0000-0002-8936-0912

## Additional files

### Major dataset

The following previously published datasets were used:

| Author(s) | Year | Dataset title | Dataset ID and/or URL | Database, license, and accessibility information |
|---|---|---|---|---|
| Foroughi LM, Kang YN, Matzger AJ | 2011 | The crystal structure of the P212121 form of bovine liver catalase previously characterized by electron microscopy | http://www.pdb.org/pdb/explore/explore.do?structureId=3nwl | Publicly available at RCSB Protein Data Bank. |
| Ko TP, Day J, Malkin AJ, McPherson A | 1999 | The structure of orthorhombic crystals of beef liver catalase | http://www.pdb.org/pdb/explore/explore.do?structureId=4blc | Publicly available at RCSB Protein Data Bank. |
| Purwar N, McGarry JM, Kostera J, Pacheco AA, Schmidt M | 2011 | Structural and kinetic analysis of the beef liver catalase complexed with nitric oxide | http://www.pdb.org/pdb/explore/explore.do?structureId=3rgp | Publicly available at RCSB Protein Data Bank. |

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
