## [Decision Letter]

Thank you for sending your work entitled “Structure of catalase solved by MicroED” for consideration at *eLife*. Your Research advance has been favorably evaluated by John Kuriyan (Senior editor) and 3 reviewers, one of whom is a member of our Board of Reviewing Editors.

The following individuals responsible for the peer review of your submission have agreed to reveal their identity: Richard Henderson (one of the peer reviewers).

The Reviewing editor and the other reviewers discussed their comments before we reached this decision and the Reviewing editor has assembled the following comments to help you prepare a revised submission.

This submission, a Research advance to the earlier Shi et al. paper, shows that electron diffraction can be used to determine the structure of catalase by molecular replacement from the known structure from x-ray crystallography. Does it add anything to the earlier paper on MicroED of lysozyme? Catalase is a much larger protein than lysozyme with much weaker diffraction, and it is a crystal form with a long track record for use as a test specimen for magnification calibration. The paper therefore does move the method forward, and it is in principle suitable to publish as an “advance”.

The reviewers were disappointed that the analysis was not very thorough. In particular:

1) The diffraction image (Figure 1) shows spots well past 2.8 A, but the structure was refined only to 3.2 A. Why was it not processed and refined to the diffraction limit of the crystal? A similar concern was raised by the reviewers of the original paper on lysozyme. It should be noted that the deposited crystal structures of catalase obtained by conventional X-ray crystallography were refined to substantially higher resolution.

2) The diffraction data are rather incomplete (∼80%), in a systematic way that by itself diminishes accuracy (missing wedge).

3) The omit maps (Figure 3) are poor, even considering a refinement-diffraction limit of 3.2 A. The authors need to determine why these maps are so poor. Is it related to data quality? They may wish to use the X-ray diffraction data of catalase (PDB IDs 3NWL or 4BLC), truncated to 3.2 A, and then perform similar validation tests, as a benchmark. It certainly would have been nearly impossible to identify the NADP molecule in such a poor map (Figure 3), and the heme group would have been a stretch, even knowing the identity of the ligand.

4) Only a single data set was analyzed, Redundancy could have improved the quality of the data, and it would have allowed the authors to assess the limitations imposed by data accuracy.

Reviewers also raised some more specific technical questions:

1) In the legend to Figure 1, the crystal shown is not the crystal (size roughly 12x4 um) used for data collection. The electron diffraction pattern is not the one used to collect the data used in the structure determination. The legend and the text state that the crystals were 8x4 um on average, yet the crystal used for the data collection was one of the larger ones at 15x7 um, and at 0.18um thick was also one of the thicker ones. In a sense, this rather undermines the unique selling point of MicroED. This observation does not invalidate the paper, but does reduce its potential impact. On the other hand, the work presented does demonstrate that there are no other serious problems (such as dynamical scattering); if it works on a big crystal, then it should also work on a smaller crystal, although it will be more work to collect the data because more crystals would be needed.

2) It is stated that an average crystal shows a maximum intensity for the strongest spot of 8.4 x 10**-3 of the direct beam. This is about 1% so the amplitude of that diffracted beam would be about 10% of the direct beam. This would produce some dynamical scattering. Since the crystal they actually used for the data collection was bigger than the average, it would presumably have a larger contribution from dynamical scattering. It would be useful if the authors had tried to further refine the structure using a protocol such as used by Grigorieff & Henderson in Ultramicroscopy, 1996. This would show how much the R-factor had been affected by dynamical scattering (i.e. increased by how much).

There was some discussion among the reviewers about whether these weaknesses eliminated the submission from further consideration, but we ultimately concurred that with careful attention to the points above, this work would meet the criteria of the new *eLife* category of “advance”. We would thus be willing to review a revised version that met the following criteria:

1) Analysis at full resolution. In the current version, the authors write that they cut off the data at 3.2 A because of merging statistics (text,Table 1). But an Rmerge of 32.7% and I/sigma(I) of 2 in the highest resolution bin is too strict. The authors should go at least to I/sigma(I) ∼ 1 and Rmerge in the last bin of 50% or even higher.

2) Efforts to reduce the substantial missing wedge (incompleteness).

3) Explicit evaluation of data quality and the effect of data quality on the clarity of the omit maps.

4) Evaluation of data from multiple crystals (this could be part of the response to criteria 2 and 3).

Additional analyses that the authors might wish to carry out (but not required for review) include use of several smaller and thinner crystals for data collection and a quantitative effort to get a lower R-factor by an explicit attempt to model the dynamical scattering quantitatively.

[Editors' note: further revisions were requested prior to acceptance, as described below.]

Thank you for resubmitting your Research advance entitled “Structure of catalase solved by MicroED” for further consideration at *eLife*. Your revised article has been favorably evaluated by John Kuriyan (Senior editor), a Reviewing editor, and two reviewers (Axel T Brunger and Richard Henderson). The manuscript has been improved but there are some remaining issues that need to be addressed before acceptance, as outlined below:

The reviewers agreed that many of their questions were answered by the revisions, although at least one of them felt that the paper would have been better had the authors collected more data from more crystals. Despite some mild disappointment, the consensus was to recommend publication, if the authors comply with the requests that follow.

1) The section that describes how the authors collected diffraction data from five different crystals is useful, since it shows how merging the data did not result in any improvement. They ascribe this outcome to lack of isomorphism, presumably due to slight cell dimension changes during the freezing. The authors should include in Figure 1 what the cell dimensions were for each of the five crystals, to show whether the non-isomorphism is apparent as cell dimension variability.

2) A useful addition is comparison of the molecular replacement solution and Fo-Fc difference maps with those from a set of X-ray data truncated to the same (3.2 Å) resolution. A “MOLREP contrast score” of 35.8 and 40.8 for the electron diffraction and equivalent truncated X-ray data will not be understood by most readers, however, as few of them will have used MOLREP. The authors should give a reference to an earlier paper or to a review by Vagin & Teplyakov in which the significance of 35.8 or 40.8 is discussed.

3) The reasons for data truncation at 3.2 Å should be stated clearly again (i.e., that the tilted crystals did not yield data beyond that limit).

4) Figure 4–figure supplement 2. This entire supplemental figure is an important validation of the method, so it must be included in the main part of the manuscript. Moreover, two different contour levels must be provided in panels C and D – it appears that the synchrotron map is better connected, whereas the microED map shows a false connection for the NADP. These differences will become probably become even clearer at a lower contour level.

5) In connection with the omit map results, currently in Figure 4–figure supplement 2 (to be moved to main part of the manuscript), it should be stated clearly in the main text that at the current level of the microED methodology, the maps obtained by microED are not of the same quality as those obtained by x-ray diffraction at synchrotrons. The microED maps are certainly not of “similar quality”, as stated (incorrectly) in the text.

---

## [Author Response]

*1) The diffraction image (*Figure 1*) shows spots well past 2.8 A, but the structure was refined only to 3.2 A. Why was it not processed and refined to the diffraction limit of the crystal? A similar concern was raised by the reviewers of the original paper on lysozyme. It should be noted that the deposited crystal structures of catalase obtained by conventional X-ray crystallography were refined to substantially higher resolution*.

As presented in the revised manuscript, we have processed the data to 2.8 Å resolution. However, we note that such high resolution was obtained from untilted crystals. Once the crystals were tilted the resolution was typically lower. Therefore the data completeness at high resolution was rather low and we decided to truncate the resolution to a final 3.2 Å.

*2) The diffraction data are rather incomplete (∼80%), in a systematic way that by itself diminishes accuracy (missing wedge)*.

The reviewers are correct in that this data does suffer from a missing wedge. Catalase crystals are long and wide but are thin and they always orient on the grid with their c-axis parallel to the electron beam. As the reviewers know, one can only tilt to a maximum angle of ∼70° in the EM before the beam is blocked by the holder. Therefore, crystals such as these will always have a missing wedge and are analogous to traditional 2D crystals in that respect.

We note that for different crystal forms, for example lysozyme, as crystals randomly orient on the grid the addition of multiple data sets from different crystals reduces and can eliminate the missing wedge. Therefore this problem is not universal but unfortunately does exist for catalase.

As we tilted the crystals up to 60° the expected missing wedge is ∼15% (15) while our data suggests ∼80% completeness. Therefore we have covered the reciprocal space as much as we could and additional data from multiple crystals did not increase the completeness. We present this data in the revised version.

Nevertheless, all of our map and model validations indicate that the final map is of good quality that does not suffer from major deformations (for example stretching of the maps as observed in some examples of density maps from 2D crystals).

*3) The omit maps (*Figure 3*) are poor, even considering a refinement-diffraction limit of 3.2 A. The authors need to determine why these maps are so poor. Is it related to data quality? They may wish to use the X-ray diffraction data of catalase (PDB IDs 3NWL or 4BLC), truncated to 3.2 A, and then perform similar validation tests, as a benchmark. It certainly would have been nearly impossible to identify the NADP molecule in such a poor map (*Figure 3*), and the heme group would have been a stretch, even knowing the identity of the ligand*.

We thank the reviewers for this suggestion. We have now generated maps using the published X-ray data (PDB ID: 3NWL) truncated to 3.2 Å and perform a comparison with our density maps for the NADP. The analysis suggests that at this resolution the density map from X-ray is equivalent in quality to the density map from MicroED. Moreover, we performed additional analyses that are presented in this revised manuscript. Phasing with a poly-alanine model indicated that the density “mutates” toward the correct residues (Figure 1). Finally, we subjected the initial resulting maps (prior to refinement) to auto ligand find and auto build. The first found strong match for 2 of the 4 heme groups and the latter was able to build ∼65% of the protein automatically. These results indicate that the catalase data are of high quality.

All of these tests have been included in the revised text.

*4) Only a single data set was analyzed, Redundancy could have improved the quality of the data, and it would have allowed the authors to assess the limitations imposed by data accuracy*.

Please see discussion above. We now include an analysis of 5 crystals and show that since all catalase crystals orient the same way on the grid with their c-axis parallel to the electron beam, we will always have a missing wedge of data in the catalase example. Therefore data from additional crystals did not increase the completeness. This discussion is now included in the revised text.

Reviewers also raised some more specific technical questions:

*1) In the legend to*
Figure 1*, the crystal shown is not the crystal (size roughly 12x4 um) used for data collection. The electron diffraction pattern is not the one used to collect the data used in the structure determination. The legend and the text state that the crystals were 8x4 um on average, yet the crystal used for the data collection was one of the larger ones at 15x7 um, and at 0.18um thick was also one of the thicker ones. In a sense, this rather undermines the unique selling point of MicroED. This observation does not invalidate the paper, but does reduce its potential impact. On the other hand, the work presented does demonstrate that there are no other serious problems (such as dynamical scattering); if it works on a big crystal, then it should also work on a smaller crystal, although it will be more work to collect the data because more crystals would be needed*.

We believe that there is a misunderstanding. Even though the crystals are wide and long, they are still very thin at 100-200 nm. It is the thickness of the crystal, through which the electron beam must pass, that could be the limiting factor when considering dynamical scattering, not the width and length of the crystal. Please remember that we cannot tilt the crystals completely on their side so the beam never travels through a 15 micron or 7 micron thick crystal. Moreover, we use a selected area aperture (6 micron diameter at the specimen), therefore we disagree that this “undermines the unique selling point of MicroED” because crystals of 100-200 nm thickness cannot be studied by traditional x-ray crystallography but are studied here.

*2) It is stated that an average crystal shows a maximum intensity for the strongest spot of 8.4 x 10**-3 of the direct beam. This is about 1% so the amplitude of that diffracted beam would be about 10% of the direct beam. This would produce some dynamical scattering. Since the crystal they actually used for the data collection was bigger than the average, it would presumably have a larger contribution from dynamical scattering. It would be useful if the authors had tried to further refine the structure using a protocol such as used by Grigorieff & Henderson in Ultramicroscopy, 1996. This would show how much the R-factor had been affected by dynamical scattering (i.e. increased by how much)*.

We thank the reviewers for this fine suggestion.

We have shown previously that an error of up to 35% in intensity values still yield acceptable data (26). Others have also shown that for catalase crystals of this thickness, kinematic scattering can be assumed (31; 7). Even though the crystal we chose to focus on is one of the thicker ones in our sample, it is still only ∼180 nm in thickness and kinematic scattering can still be assumed as we discussed.

We agree, however, with the reviewers that some dynamical scattering will be produced and we have not, as of yet, applied any corrections. We plan to do so in the future.

*There was some discussion among the reviewers about whether these weaknesses eliminated the submission from further consideration, but we ultimately concurred that with careful attention to the points above, this work would meet the criteria of the new* eLife *category of “advance”*. *We would thus be willing to review a revised version that met the following criteria:*

*1) Analysis at full resolution. In the current version, the authors write that they cut off the data at 3.2 A because of merging statistics (text,*Table 1*). But an Rmerge of 32.7% and I/sigma(I) of 2 in the highest resolution bin is too strict. The authors should go at least to I/sigma(I) ∼ 1 and Rmerge in the last bin of 50% or even higher*.

We now include analysis at full resolution and from multiple crystals. We have presented a more detailed analysis on how the final data set was chosen based on completeness, merging and the final refinement statistics. In short, the low information content in the higher resolution shells precludes their inclusion.

*2) Efforts to reduce the substantial missing wedge (incompleteness)*.

As discussed above, the preferred orientation of the catalase crystals on the grid means that one will always suffer from a missing wedge with this sample. Nevertheless, we now present data from multiple crystals and explain that because of the preferred orientation of the crystals, merging multiple data sets together does not improve the data completeness over the completeness of crystal 4 dataset alone. Tilting the TEM stage beyond ∼60° for this sample (the angle used for the single crystal data set) did not provide useful data as the diffraction was generally quite poor at such high tilts.

*3) Explicit evaluation of data quality and the effect of data quality on the clarity of the omit maps*.

We have provided more detailed analysis in the Model validation section, specifically the results of autobuilding, automated ligand finding, and comparisons with X-ray data as suggested by the reviewers.

*4) Evaluation of data from multiple crystals (this could be part of the response to criteria 2 and 3)*.

Done. As mentioned above we added our analysis on the effects of using multiple crystals.

*[Editors' note: further revisions were requested prior to acceptance, as described below*.*]*

*1) The section that describes how the authors collected diffraction data from five different crystals is useful, since it shows how merging the data did not result in any improvement. They ascribe this outcome to lack of isomorphism, presumably due to slight cell dimension changes during the freezing. The authors should include in*
Figure 1
*what the cell dimensions were for each of the five crystals, to show whether the non-isomorphism is apparent as cell dimension variability*.

We have added the unit cell dimensions for the five crystals in Supplementary file 1. We also added text explaining that four of the crystals were relatively isomorphous; however, even when combined they yielded relatively incomplete data.

*2) A useful addition is comparison of the molecular replacement solution and Fo-Fc difference maps with those from a set of X-ray data truncated to the same (3.2 Å) resolution. A “MOLREP contrast score” of 35.8 and 40.8 for the electron diffraction and equivalent truncated X-ray data will not be understood by most readers, however, as few of them will have used MOLREP. The authors should give a reference to an earlier paper or to a review by Vagin & Teplyakov in which the significance of 35.8 or 40.8 is discussed*.

We thank the reviewers for this suggestion. We have provided a contrast score for an MR search with X-ray data and have provide what is considered a high quality score (>3.0) as defined in the official CCP4 documentation for MOLREP. We have also included omit maps generated from X-ray data truncated to 3.2 Å for comparison and added them to what is now called Figure 3.

*3) The reasons for data truncation at 3.2 Å should be stated clearly again (i.e., that the tilted crystals did not yield data beyond that limit)*.

This has been added to the text.

*4) Figure 4–figure supplement 2. This entire supplemental figure is an important validation of the method, so it must be included in the main part of the manuscript. Moreover, two different contour levels must be provided in panels C and D – it appears that the synchrotron map is better connected, whereas the microED map shows a false connection for the NADP. These differences will become probably become even clearer at a lower contour level*.

Supplemental Figure 2 has been moved back into the main manuscript and renamed Figure 3. Two different contour levels for the NADP have been added to this figure as requested.

*5) In connection with the omit map results, currently in Figure 4–figure supplement 2 (to be moved to main part of the manuscript), it should be stated clearly in the main text that at the current level of the microED methodology, the maps obtained by microED are not of the same quality as those obtained by x-ray diffraction at synchrotrons. The microED maps are certainly not of “similar quality”, as stated (incorrectly) in the text*.

It was never the focus of this work to directly compare the quality of maps obtained by MicroED with those obtained by X-ray diffraction. Regardless, we have added X-ray omit maps for comparison and have stated that the MicroED maps are not of the same quality. However, it needs to be noted that traditional X-ray crystallography could have never provided data, and subsequently density maps, from the crystals used in this study, as they are much too small.